# Mitoquinone Alleviates Donation after Cardiac Death Kidney Injury during Hypothermic Machine Perfusion in Rat Model

**DOI:** 10.3390/ijms241914772

**Published:** 2023-09-30

**Authors:** Anna Radajewska, Jakub Szyller, Anna Krzywonos-Zawadzka, Agnieszka Olejnik, Grzegorz Sawicki, Iwona Bil-Lula

**Affiliations:** 1Division of Clinical Chemistry and Laboratory Hematology, Department of Medical Laboratory Diagnostics, Faculty of Pharmacy, Wroclaw Medical University, Borowska 211A, 50-556 Wroclaw, Poland; anna.radajewska@umw.edu.pl (A.R.); jakub.szyller@umw.edu.pl (J.S.); agnieszka.olejnik@umw.edu.pl (A.O.); grzegorz.sawicki@umw.edu.pl (G.S.); 2Department of Anatomy, Physiology and Pharmacology, College of Medicine, University of Saskatchewan, Saskatoon, SK S7N 5E5, Canada

**Keywords:** kidney graft, hypothermic machine perfusion, MitoQ, ischemia–reperfusion injury, oxidative stress

## Abstract

Transplanted organs are subjected to harmful conditions through stopping blood flow, hypothermic storage of the graft, and subsequent reperfusion. In particular, kidneys donated from patients after cardiac arrest (DCD) are classified as more vulnerable to ischemia–reperfusion injury (IRI). Hypothermic machine perfusion is proposed as a solution for better kidney storage before transplantation, and it is a good platform for additional graft treatment. Antioxidants have gained interest in regenerative medicine due to their ability to scavenge reactive oxygen species (ROS), which play a key role in IRI. We evaluated the effect of Mitoquinone (MitoQ), a strong mitochondria-targeted antioxidant, administered directly to the perfusing buffer. Rat kidneys were isolated, randomly classified into one of the following groups, donation after brainstem death (DBD), DCD, and DCD with MitoQ, and perfused for 22 hours with a hypothermic machine perfusion system. Subsequently, we detected levels of kidney injury (KIM-1) and oxidative stress (ROS/RNS, cytochrome C oxidase, and mitochondrial integrity) markers. We compared the activation of the apoptosis pathway (caspase 3 and 9), the concentration of phosphorylated Akt (pAkt), and the pAkt/total Akt ratio. MitoQ reduces KIM-1 concentration, total ROS/RNS, and the level of caspases. We observed a decrease in pAkt and the pAkt/total Akt ratio after drug administration. The length of warm ischemia time negatively impacts the graft condition. However, MitoQ added to the perfusing system as an ‘on pump’ therapy mitigates injury to the kidney before transplantation by inhibiting apoptosis and reducing ROS/RNS levels. We propose MitoQ as a potential drug for DCD graft preconditioning.

## 1. Introduction

End-stage renal disease (ESRD) is the final stage of chronic kidney disease (CKD) that needs an urgent intervention, of which kidney transplantation is preferred. Compared to other therapeutic opinions, both hemodialysis and peritoneal dialysis, kidney transplantation increases the patient’s quality of life and reduces the mortality rate [1]. Multiple countries are allowed to use expanded criteria donor (ECD) kidneys in order to increase the pool of kidney grafts suitable for donation. Despite the greater number of possible graft donors, there arises a problem because ECD or kidneys donated after a cardiovascular cause of death (DCD) are more vulnerable to ischemia–reperfusion injury (IRI), and consequently primary nonfunction (PNF) or delayed graft function (DGF) [2].

IRI that occurs right after cardiac arrest, called warm ischemia time (WIT), continues during organ preservation, known as cold ischemia time (CIT) [3]. Reperfusion injury is related to the restoration of blood flow and paradoxically causes strong damage through oxidative stress and massive reactive oxygen species production [4]. IRI reduces the quality of a kidney transplant and is related to delayed graft function, which requires repeating a dialysis treatment [5]. In addition, transplanted kidneys are at risk of damage due to a reduction in active nephrons, acute injury, and constant toxic injury caused by immunosuppressive drugs. Mechanical perfusion is a recently used solution for DCD kidneys, where the graft is kept in a hypothermic condition and the flow of the buffer provides vascular patency and excises the toxic wastes [6]. Moreover, it is a perfect solution for additional treatment ‘on pump’ during the unavoidable time of running the surgical procedures. Therefore, preconditioning a kidney graft during machine perfusion is one of the possible ways to improve the kidney’s condition. 

Oxidative stress is defined as an imbalance between the generation of free radicals or oxidants and the intercultural antioxidative mechanisms that reduce the highly reactive particles [4]. Recently, the adverse effect of oxidants has been thoroughly studied because it is proposed as an important factor for the reactivity of the immune system in diseases like acute and CKD, cardiovascular disease, diabetes, and metabolic syndrome [7]. CKD and dialysis are associated with elevated levels of oxidative stress markers and pro-inflammatory cytokines [8,9]. The whole process of kidney transplantation, starting with surgical intervention, organ storage, reperfusion, and further immunosuppressive treatment, causes massive production of oxidants and insufficient efficiency of the antioxidant system [10,11]. Oxidative stress induces DNA fragmentation and cytochrome C release that trigger the activation of apoptosis [12]. La Manna et al. showed that a low oxidation level and apoptosis 6 months after kidney transplantation correlate with better functioning of the kidney graft. Moreover, they noticed a correlation between IL-10 level and DNA oxidation, and between both IL-6 and IL-10 and the level of fragmented DNA [13]. 

In this study, we compared the differences in oxidative stress and apoptosis activation between donation after brainstem death (DBD) and DCD kidney grafts subjected to 22 h of hypothermic machine perfusion. Furthermore, we evaluated Mitoquinone (MitoQ) organ preconditioning. We proposed to use a mitochondria-targeted antioxidant, MitoQ, with high antioxidative properties. MitoQ is a lipophilic cation that easily penetrates the mitochondrial membrane barrier and accumulates internally [14]. The additional advantage of MitoQ is the ubiquinone moiety in the structure, which is activated by Complex II to the ubiquinol, reduces ROS, and is recycled back to the form that can be reused, which provides a high antioxidant capacity [14]. Mitoquinone mesylate was previously clinically tested for the treatment of hepatitis C [15], non-alcoholic fatty liver disease [16], and insulin resistance [17]. Nowadays, a few clinical studies have been performed: for antiviral treatment, dilated cardiomyopathy [18], and diastolic dysfunction [19]. MitoQ was also shown to reduce the oxidative damage associated with clinical acute kidney injury (AKI) [20]. Therefore, we chose MitoQ as a good candidate for ameliorating ischemia–reperfusion injury, which is strongly associated with oxidative stress.

## 2. Results

### 2.1. The Influence of MitoQ Administration on the Level of Kidney Injury Markers

KIM-1 is known as a marker of kidney injury. We showed elevated concentrations of KIM-1 in the DCD group (113.51 ± 63 ng/mL vs. 260.03 ± 40.72 ng/mL, *p* = 0.012 for DBD vs. DCD) compared to DBD. The administration of the MitoQ into the perfusate buffer reduced the level (260.03 ± 40.72 ng/mL vs. 117.10 ± 25.49 ng/mL, *p* = 0.033 for DCD vs. DCD + MitoQ) (Figure 1).

### 2.2. Oxidative Status of Perfused Kidneys

Total ROS/RNS concentrations in kidney tissue homogenates were determined to evaluate the intensity of oxidative stress after the hypothermic machine perfusion of the rat kidneys. We found that the quantity of ROS/RNS was elevated in DCD grafts in comparison to DBD grafts (17.53 ± 7.10 nM/µg protein vs. 52.73 ± 13.63 nM/µg protein, *p* = 0.0368 for DBD vs. DCD). The group with MitoQ had significantly reduced ROS/RNS levels compared to the DCD group (52.73 ± 13.63 nM/µg protein vs. 10.97 ± 2.84 nM/µg protein, *p* = 0.0134 for DCD vs. DCD + MitoQ) (Figure 2).

### 2.3. Caspase 3 and Caspase 9 Concentrations Are Reduced after MitoQ Administration

Caspase 9 initiates the apoptosis process that further activates caspases 3, 6, or 7, known as apoptosis-executing caspases [21]. Here, we observed the increased concentration of caspase 9 (A) and caspase 3 (B) in the DCD group compared to DBD (casp 9: 7.31 ± 0.43 ng/mg protein vs. 14.86 ± 2.61 ng/mg protein, *p* = 0.028 for DBD vs. DCD; casp 3: 15.86 ± 1.23 ng/mg protein vs. 39.07 ± 5.76 ng/mg protein, *p* = 0.002 for DBD vs. DCD). MitoQ significantly reduced both initialing and executing caspase concentrations in kidney tissue (casp 9: 14.86 ± 2.61 ng/mg protein vs. 7.15 ± 1.53 ng/mg protein, *p* = 0.019 for DCD vs. DCD + MitoQ; casp 3: 39.07 ± 5.76 ng/mg protein vs. 14.82 ± 2.81 ng/mg protein *p* = 0.002 for DCD vs. DCD + MitoQ) (Figure 3).

### 2.4. MitoQ Mitigate the Ratio of Phosphorylated Akt and Total Akt in Kidney Tissue

An ELISA assay was performed to assess the concentration of phosphorylated Akt (p-Akt) in kidney tissue. The concentration was significantly higher in the DCD group compared to the DBD group (0.15 ± 0.03 ng/mg protein vs. 0.42 ± 0.06 ng/mg protein, *p* = 0.005 for DBD vs. DCD) (a). Moreover, MitoQ administration resulted in p-Akt downregulation (0.42 ± 0.06 ng/mg protein vs. 0.20 ± 0.05 ng/mg protein, *p* = 0.02 for DCD vs. DCD + MitoQ). Subsequently, we calculated the p-Akt to total Akt (tAkt) ratio (b) to show the relative changes of the active form for DBD and DCD kidney grafts, and the impact of MitoQ treatment. Here, we observed the same dependence: the p-Akt/tAkt ratio was higher for DCD grafts (1.48 ± 0.33 arbitrary units vs. 3.72 ± 0.69 arbitrary units, *p* = 0.014 for DBD vs. DCD), and the significant depletion was noticed after MitoQ supplementation (3.72 ± 0.69 arbitrary units vs. 1.79 ± 0.41 arbitrary units, *p* = 0.035 for DCD vs. DCD + MitoQ) (Figure 4). 

### 2.5. MitoQ Mitigate the Ratio of Phosphorylated Akt and Total Akt in Kidney Tissue

We did not observe significant differences for cytochrome c oxidase activity (CcO) (A) between DBD and DCD groups (0.04 ± 0.01 Units/mL vs. 0.15 ± 0.05 Units/mL, *p* = 0.091 for DBD vs. DCD) and between the DCD group and the DCD + MitoQ group (0.15 ± 0.05 Units/mL vs. 0.09 ± 0.04 Units/mL, *p* = 0.391 for DCD vs. DCD + MitoQ). We found a slight increase in CcO activity for the DCD group compared to the DBD group, but Mitoquinone administration had no effect on CcO activity. Moreover, we evaluated the mitochondrial outer membrane integrity (B), and this result did not confirm the positive effect of MitoQ administration on the percentage of mitochondria with an undamaged outer membrane (55.71 ± 11.82% vs. 68.60 ± 11.24%, *p* = 0.728 for DBD vs. DCD; 68.60 ± 11.24% vs. 29.43 ± 14.18%, *p* = 0.115 for DCD vs. DCD + MitoQ) (Figure 5).

## 3. Discussion

Mitoquinone was previously studied in different animal models of kidney transplantation [22] and tested in clinical trials, demonstrating its safety [23]. However, none of these studies combined long-term hypothermic perfusion (22 h) and the model of kidneys donated after cardiovascular causes of death with prolonged WIT. We focused on the comparison of DBD and DCD rat kidney grafts and the positive effect of MitoQ on DCD grafts. Direct MitoQ administration into the perfusion buffer helps to avoid different drug interactions and poorer MitoQ bioavailability when given orally or intravenously [24]. Additionally, kidney preconditioning might improve graft viability and minimize patient burden. This hypothesis has been illustrated in Figure 6.

In the present study, we showed that Mitoquinone, a mitochondria-targeted antioxidant, reduces ROS/RNS production and prevents the increased apoptosis of kidney cells during the ischemic phase of the IRI of transplanted kidneys. Moreover, we detected decreased concentrations of phosphorylated Akt and the pAkt/tAkt ratio after drug application. We detected an increased level of the early injury marker KIM-1 when the warm ischemia time was longer (like in DCD grafts) compared to DBD grafts, where the warm ischemia time is minimal. The preconditioning of the kidney graft with MitoQ reduced the concentration of the injury marker in DCD kidneys. Our results suggest that Mitoquinone is a possibly good candidate for preconditioning a perfused kidney graft before surgery.

Mitochondria are the main source of energy and reactive oxygen species production. ROS are important in cell signaling, but they also play a significant role in numerous diseases and the aging process [25]. Previous studies have shown that ROS are massively produced during prolonged cold storage of the organ, and during the reperfusion stage [26]. Moreover, the analysis of perfusates collected after human DCD and DBD kidney graft perfusion showed elevated injury markers (NGAL and LDH) for DCD compared to DBD [27]. Our study also proved that longer warm ischemia associated with DCD grafts resulted in increased KIM-1 protein concentrations in kidney tissue. Moreover, 30 min of WIT resulted in a higher ROS/RNS concentration compared to kidneys with minimal WIT. We detected the positive effect of MitoQ administration on injury and oxidative stress marker levels. Increased oxidative stress causes cell apoptosis through the opening of the mitochondrial permeability transition pore (PTP), and subsequently the leakage of cytochrome c, which activates the caspase-dependent apoptosis pathway [28].

In our study, we detected a significant reduction in the concentration of the important apoptosis proteins caspase 3 and caspase 9 when MitoQ was added. Mitoquinone was proven to protect gentamicin-treated kidneys from increased cell death via the depletion of caspase 3 and 7 activity [29]. The increased mitochondrial ROS production correlates with organ fibrosis and the activation of apoptosis [30]. MitoQ administration reduces the expression of liver fibrosis proteins such as Col1A1 and the active form of caspase 3, which represents the executive apoptosis protease [30].

PI3K/Akt signaling plays an important role in redox balance, apoptosis activation, and autophagy [31,32]. Akt is the serine/threonine kinase that controls various cell biology pathways, such as apoptosis, proliferation, autophagy, and metabolism. The active form of Akt probably has a dual effect on pathologic pathways and disease progression. Although Akt is proposed as an anti-apoptotic signaling molecule [33] that inhibits caspase 9 and caspase 3 activation [34], and therefore promotes cell survival, it is also suggested that high Akt activity results in disease progression. Moreover, Akt contributes to kidney fibrosis in CKD [35]. CKD is related to cell proliferation and the accumulation of extracellular matrix deposits in the glomeruli and the interstitial space. The transforming growth factor beta 1 (TGF-β1) is proposed as an important inducer of kidney fibrosis and an activator of the Akt protein [36,37]. Here, we detected a higher ratio of pAkt to total Akt connected to an increased level of the activated form in the DCD group compared to the DBD group. A long period of warm ischemia is associated with a higher phosphorylation of Akt. Moreover, MitoQ administration significantly downregulated the pAkt/total Akt ratio. The hyperactivation of Akt contributes to DNA damage and cell proliferation in polycystic kidney disease [38]. Conduit et al. propose Akt inhibitors as good candidates for PKD treatment [38].

Cytochrome C oxidase (CcO) is an important transmembrane protein located in the mitochondrion’s membrane. CcO is involved in ATP production and is part of the respiratory electron transport chain known as complex IV. Electrons are transferred from reduced cytochrome C to cytochrome C oxidase, which transfers them into molecular oxygen that is reduced to water [39,40]. The activity of CcO changes with pathologies. During brain ischemia–reperfusion, CcO becomes hyperactive and dramatically drops as mitochondrial injury progresses in the late reperfusion phase [40]. Ischemia is caused by oxygen and, therefore, energy depletion, which results in calcium accumulation, which provokes the dephosphorylation of mitochondrial proteins. The calcium signaling pathway highly activates the ETC complex, and during the reperfusion phase it demonstrates hyperpolarization of the mitochondrial membrane potential (ΔΨm) and massive ROS production [41]. However, the study of See et al. did not confirm any significant changes in CcO activity during different times of heart cold storage [42]. We did not detect any significant change in CcO activity between DBD and DCD kidneys. In addition, MitoQ administration did not have an impact on this enzyme’s activity. We also did not confirm the positive effect of the mitochondria-targeted drug on mitochondrial membrane integrity.

## 4. Material and Methods

### 4.1. Reagents

All the reagents used in this work are listed in Table 1.

### 4.2. Experimental Animals

Adult, pathogen-free, male Wistar rats weighing 300–350 g were purchased from the Mossakowski Medical Research Center, Polish Academy of Sciences, Warsaw, Poland. Two animals per cage were kept under a controlled temperature (22 + 2 °C), humidity (55% +/− 10%), light/dark cycle (12/12 h), and with unlimited food and water provided. All animal experimental procedures were approved by the local Ethics Committee for Experiments on Animals at the Ludwik Hirszfeld Institute of Immunology and Experimental Therapy, Polish Academy of Sciences, Wroclaw, Poland, and carried out in accordance with the Guide of the Polish Ministry of Science and Higher Education for the Care and Use of Experimental Animals. Animals were randomly categorized into study groups (7 animals per group).

### 4.3. Hypothermic Machine Perfusion of Rat Kidney

Rats were desensitized with buprenorphine (0.05 mg/kg, i.p.) and anesthetized with sodium pentobarbital (0.5 mL/kg i.p.). Then, the heart was excised from the animal, which initiated warm ischemia. The right kidney was immediately excised, cannulated by the renal artery, and connected to the hypothermic machine perfusion system (Single Channel Perfusion System, EMKA Technologies, Paris, France) (DBD group) or tied for 30 min with controlled temperature (37 °C) to mimic DCD graft donation and subsequently excised, cannulated by the renal artery, and connected to the hypothermic machine perfusion system (experimental groups: DCD and DCD + MitoQ) (Figure 7). Kidneys were perfused for an hour with a perfusion buffer to wash out blood cells, and then disposed of and replaced with 100 mL of fresh perfusion buffer with Mitoquinone (DCD + MitoQ group) or without MitoQ (DBD and DCD group). The final MitoQ concentration was 18 µmol/l. It was selected based on the safe and effective dose studied by Dare et al. on a mouse renal ischemia–reperfusion injury model (in vivo study) [20]. The perfusion buffer was based on Krebs–Henseleit buffer (pH = 7.4; 118 mmol/L NaCl, 4.7 mmol/L KCl, 1.2 mmol/L KH_2_PO_4_, 1.2 mmol/L MgSO_4_, 3.0 mmol/L CaCl_2_, 25 mmol/L NaHCO_3_, 11 mmol/L glucose, and 0.5 mmol/L EDTA) with albumin. The kidney was perfused with cold buffer (4 °C) for 22 h with a constant flow rate (0.12 mL/min) to maintain the pressure between 50 and 120 mmHg. The output of the monitoring system was recorded using IOX2 software (EMKA Technologies, Paris, France).

### 4.4. Preparation of Kidney Homogenates

Kidneys were kept at -80℃ and subsequently crushed in liquid nitrogen. Kidney tissue was homogenized with a manual homogenizer in cold homogenization buffer (50 mmol/L Tris-HCl (pH 7.4), 3.1 mmol/L sucrose, 1 mmol/L dithiothreitol, 10 mg/mL leupeptin, 10 mg/mL soybean trypsin inhibitor, 2 mg/mL aprotinin, and 0.1% Triton X-100). The homogenates were centrifuged for 10 min at 14,000 RPM, and the supernatants were collected and stored at −80 °C.

### 4.5. Assessment of Kidney Injury Marker (KIM-1)

Kidney injury marker 1 is an early marker of kidney cell injury, especially of proximal tubule cells [43]. KIM-1 concentration in kidney homogenates was determined with an ELISA kit (cat. E0785r, EIAab Science Inc., Wuhan, China). To determine KIM-1 protein concentration, we used 100 µL of kidney homogenates diluted 1000 times. The detected concentration of KIM-1 in the sample was then normalized to the total protein concentration in the homogenate. Briefly, the primary antibodies bound to KIM-1. Then, KIM-1 was detected with anti-KIM-1 secondary antibodies conjugated with biotin and enzyme-conjugated avidin. The substrate solution was added and a color proportional to the protein concentration was detected using a Spark Multimode Microplate Reader (Tecan Group Ltd., Männedorf, Switzerland) at 450 nm.

### 4.6. Detection of Total Protein Concentration

The Bradford method was used to detect the total protein concentration in the kidney homogenate samples. Briefly, Bio-Rad Protein Assay Dye Reagent (cat. 5000006, Bio-Rad, Hercules, CA, USA) was used and BSA (heat shock fraction, ≥98%, Sigma-Aldrich, Saint Louis, MO, USA) was used as a standard. The absorbance measured at 595 nm is proportional to the protein concentration. Absorbance was measured on Spark multimode microplate reader (Tecan Trading AG, Männedorf, Switzerland).

### 4.7. Assessments of ROS/RNS in the Kidney Tissue

To evaluate the oxidative stress in kidney tissue, the OxiSelect™ In Vitro ROS/RNS Assay Kit (cat. STA-347 Cell Biolabs, San Diego, CA, USA) was used. In the presence of ROS and RNS, dichlorodihydrofluorescein DiOxyQ (DCFH-DiOxyQ) is oxidized to the fluorescent 2′,7′-dichlorodihydrofluorescein (DCF). The fluorescence signal was measured at 480 nm (excitation) and 530 nm (emission) with a Spark Multimode Microplate Reader (Tecan Trading AG, Männedorf, Switzerland). The total ROS/RNS concentration is proportional to the fluorescent signal. The detected concentration was normalized to the total protein concentration.

### 4.8. Mitochondria Isolation

Intact mitochondria were isolated from frozen kidney tissue with the MinuteTM Mitochondria Isolation Kit for Muscle Tissues/Cultured Muscle Cells (cat. MM-038, Invent Biotechnologies, Plymouth, MA, USA). This kit allows for the isolation of mitochondria, cytosol, and nuclei. Approximately 30 µg of minced tissue and dissociation beads was placed on the surface of the filter cartridge. The buffer was added, and the tissue paste was ground and centrifuged for 30 s (16,000× *g*). The filter was discarded, and the supernatant was vortexed and centrifuged for 5 min (1000× *g*). Next, the obtained supernatant was transferred to a new microtube and centrifuged for 20 min (11,000× *g*). Then, the supernatant was discarded, and the pellet was resuspended in 200 µL of the provided buffer and vortexed. The tube was centrifuged for 10 min (11,000× *g*) and the obtained supernatant was transferred to a fresh tube with ice-cold DPBS and vortexed. Subsequently, the tube was centrifuged for 20 min (16,000× *g*) to obtain the pellet of the mitochondrial fraction. Mitochondria were dissolved in 50 µL of the homogenization buffer (50 mmol/L Tris-HCl (pH 7.4), 3.1 mmol/L sucrose, 1 mmol/L dithiothreitol, 10 mg/mL, leupeptin, 10 mg/mL soybean trypsin inhibitor, 2 mg/mL aprotinin, and 0.1% Triton X-100), and the total protein concentration was measured.

### 4.9. Measurements of Cytochrome C Oxidase Activity and Outer Mitochondrial Membrane Integrity

Cytochrome C oxidase activity and outer mitochondrial membrane integrity were measured in isolated mitochondria with the Cytochrome C Oxidase Assay Kit (cat. CYTOCOX1 Sigma-Aldrich). Reduced ferrocytochrome c is oxidized in a two-phased reaction by cytochrome c oxidase with a sudden burst of activity [44]. Briefly, in the quart cuvette, we mixed 950 µL of assay buffer, 100 µL enzyme dilution buffer, and 20 µL of isolated sample. By adding 50 µL of ferrocytochrome c substrate solution, the reaction was started. The absorbance at 550 nm was kinetically measured after 5 and 60 s of the reaction. To subtract the background values, we measured the change in absorbance in a mix of 950 µL of assay buffer, 120 µL enzyme dilution buffer, and 50 µL of ferrocytochrome c substrate solution. Subsequently, CcO activity was calculated using the equation: Units/mL $=\frac{\Delta Ab/min\;\times \;dil\;\times \;1.1}{(volofenzyme)\;\times \;21.84}$ (where ΔA/min = A/minute (sample) − A/minute (blank); dil = dilution factor of enzyme or sample; 1.1 = reaction volume in ml; vol of enzyme = volume of enzyme or sample in ml; 21.84 = Δɛ^mM^ mM between ferrocytochrome c and ferricytochrome c at 550 nm). The activity of the sample was normalized based on total mitochondrial protein concentration. To determine the outer mitochondrial membrane integrity, the activity of CcO was measured in the presence and absence of n-dodecyl β-D-maltoside. To detect intact mitochondria, 0.1 mg protein/mL of mitochondrial suspension was mixed with 1x enzyme dilution buffer. For total cytochrome c oxidase activity detection, the same amount of the same sample was mixed with Enzyme Dilution Buffer containing 1 mM n-dodecyl β-D-maltoside. Both prepared samples were incubated for 10 min on ice. A total of 1 µg of mitochondrial protein was used, following the CcO activity assay described above. The calculated ratio between measurements of samples with and without detergent stands for the integrity of the outer mitochondrial membrane and is presented as a percentage of mitochondria with an undamaged outer membrane:

($\%=\frac{(\Delta A(total)\;-\;\Delta A(intact))\;\times \;100}{\Delta A(total)};\;\Delta A\left(intact\right)=\Delta A\left(intact\;sample\right)\;-\;\Delta A(blank)$; ΔA(total) = Δ*A*(total sample) − Δ*A*(blank)). The measurements were performed on a spectrophotometer (UV-Vis Double Beam HALO DB-20 Spectrofotometer, Dynamica GmbH, Zürich, Switzerland) in the Laboratory of Elemental Analysis and Structural Research of Wroclaw Medical University.

### 4.10. Caspase 3 and Caspase 9 Concentration Measurement

Caspase 3 and caspase 9 concentrations were measured in kidney homogenates according to company instructions (cat. E1648Ra, Rat Caspase 3, CASP3 ELISA kit and cat. E1898Ra, Rat Caspase 9, CASP9 ELISA kit, BT Lab, Birmingham, UK). Briefly, caspase 3 or caspase 9, respectively, binds to the pre-coated plate. A biotinylated antibody is added and binds to the protein. Streptavidin-HRP antibody is added, and, after incubation, the substrate solution demonstrates a developing color that is proportional to caspase concentration. The absorbance is then determined using a Spark multimode microplate reader (Tecan Trading AG, Switzerland) at 450 nm. We used 100 µL of kidney homogenate diluted 200 times. The results were normalized to the total protein level.

### 4.11. Total Akt and Phosphorylated Akt Concentration

Total Akt was detected using ELISA (cat. PEL-AKT-S473-T-1, RayBio^®^ Human, Mouse and Rat Phospho-AKT (Ser473) and Total AKT ELISA Kit), and the phosphorylated Akt form (Ser473) concentration was determined using the Rat Phospho-AKT(S473) ELISA Kit (cat. E2452Ra, BT Lab). Phospho-Akt concentration was normalized with the total protein level. Later, we calculated the ratio of phospo-Akt and total Akt, which better reflects the active form level than pAkt concentration.

### 4.12. Statistical Analysis

The data were collected and further analyzed using GraphPad Prism 8.0.1 for Windows (GraphPad Software, San Diego, CA, USA). The normality of variance changes was determined using the Shapiro–Wilk normality test. ANOVA or nonparametric equivalent tests were performed, and post hoc tests were used for comparisons between groups. The obtained results were expressed as mean ± SEM, and a *p* value of < 0.05 stands for significant change.

## 5. Study Limitations

Our study has a few limitations. Firstly, we did not perform the kidney transplantation procedure; therefore, we were not able to evaluate the post-surgical effect of MitoQ graft preconditioning. Secondly, due to technical limitations, we were able to detect the input pressure of the canula and maintain it around 50–120 mmHg, although different pressures might be detected inside the organ. We believe the next step should be the evaluation of MitoQ’s positive effect on kidney injury and apoptosis after the reperfusion phase, where the ROS/RNS burst occurs.

## 6. Conclusions

Our study showed that warm ischemia time might play an important role in the condition of kidney grafts that undergo hypothermic machine perfusion. We showed that a longer WIT, like for DCD kidneys, increases the levels of oxidative stress and injury markers, resulting in a higher concentration of pro-apoptotic caspases in kidney tissue. Moreover, we propose MitoQ, a mitochondria-targeted antioxidant, as a good candidate to reduce the harmful effect of prolonged WIT, and the following CIT, via a reduction in oxidative stress and apoptotic cell death.

## Figures and Tables

**Figure 1 ijms-24-14772-f001:**
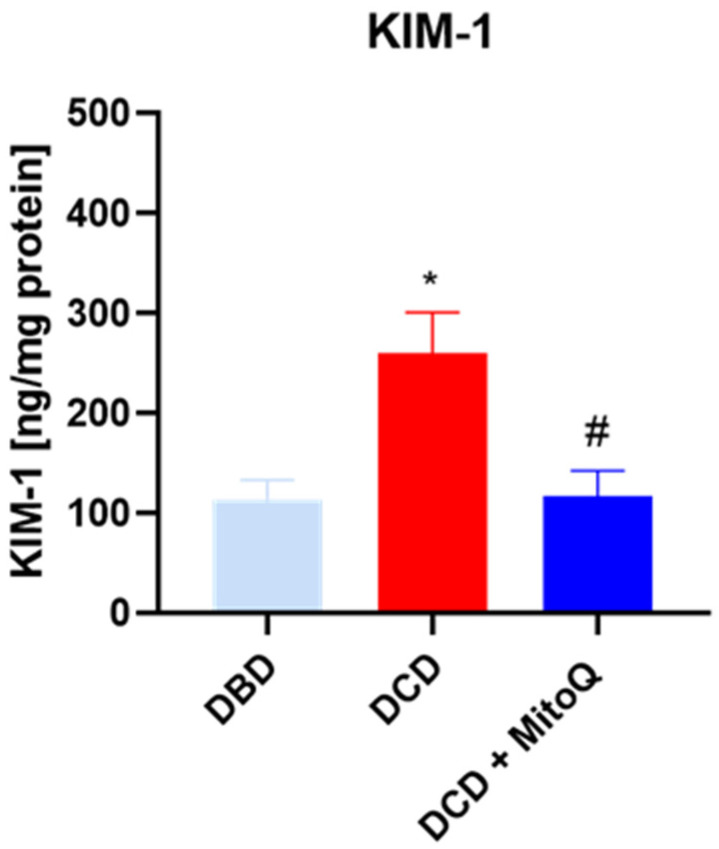
Concentration of KIM-1 in kidney homogenates (data represent mean ± SD (n = 7), * *p* < 0.05 DBD vs. DCD, # *p* < 0.05 DCD vs. DCD + MitoQ).

**Figure 2 ijms-24-14772-f002:**
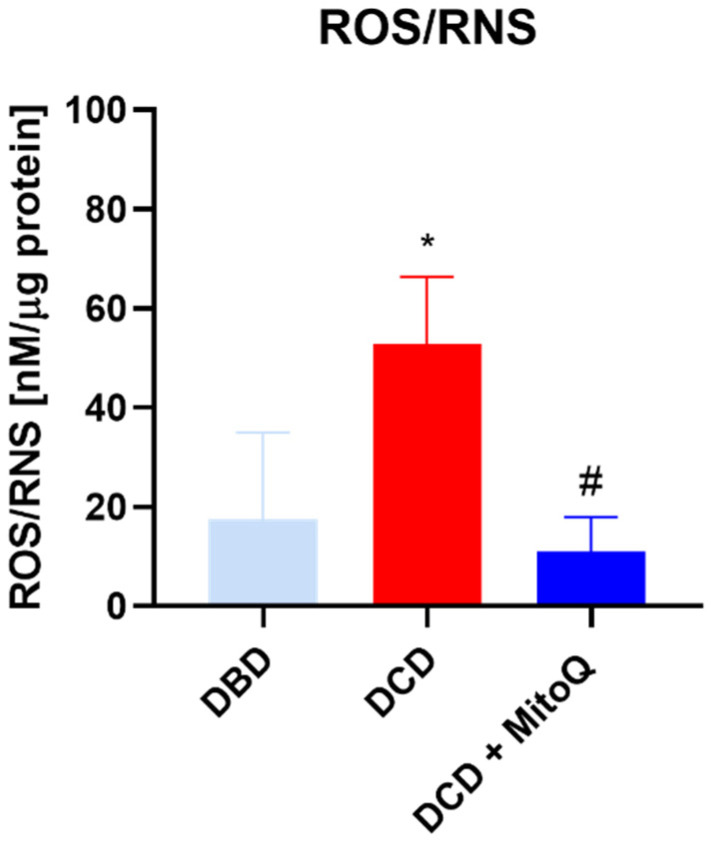
Concentration of ROS/RNS in kidney homogenates (data represent mean ± SEM (n = 6), * *p* < 0.05 DBD vs. DCD, # *p* < 0.05 DCD vs. DCD + MitoQ).

**Figure 3 ijms-24-14772-f003:**
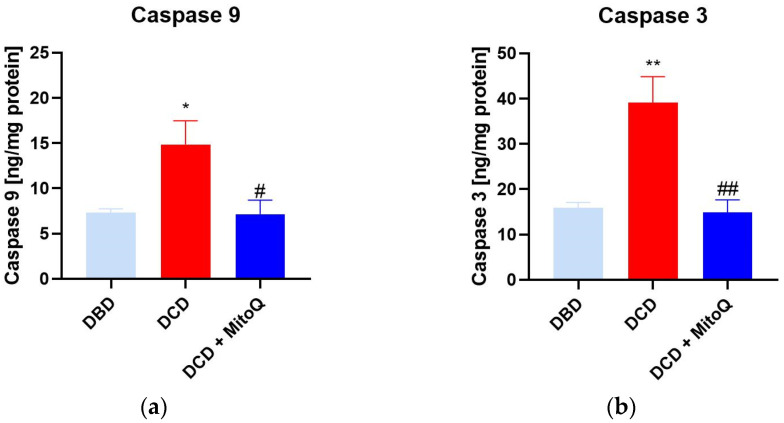
(**a**) Caspase 9 and (**b**) caspase 3 concentration in kidney tissue (data represent mean ± SEM (n = 6), * *p* < 0.05, ** *p* < 0.01 DBD vs. DCD, # *p* < 0.05, ## *p* < 0.01 DCD vs. DCD + MitoQ).

**Figure 4 ijms-24-14772-f004:**
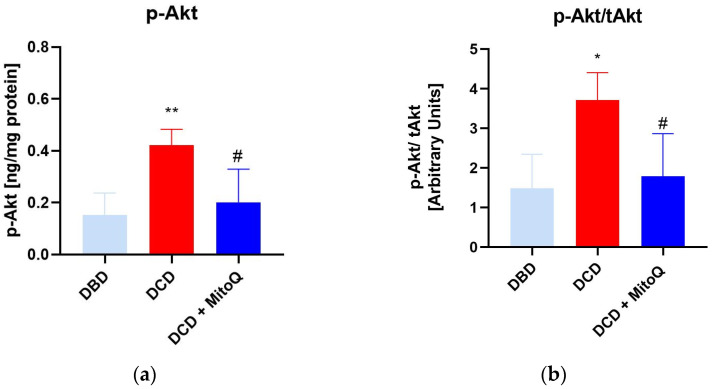
(**a**) p-Akt concentration and (**b**) ratio of p-Akt/tAkt in kidney tissue (data represent mean ± SEM (n = 7), * *p* < 0.05, ** *p* < 0.01 DBD vs. DCD, # *p* < 0.05 DCD vs. DCD + MitoQ).

**Figure 5 ijms-24-14772-f005:**
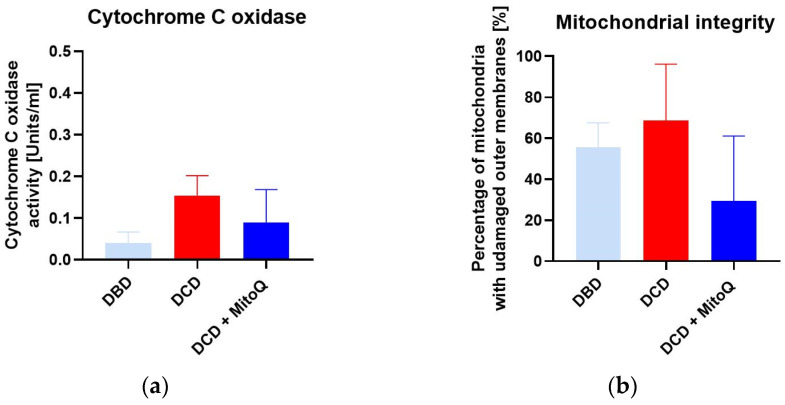
(**a**) Cytochrome C oxidase activity (CcO) and (**b**) mitochondrial outer membrane integrity in isolated mitochondria from kidney tissue (data represent mean ± SEM (n = 6)).

**Figure 6 ijms-24-14772-f006:**
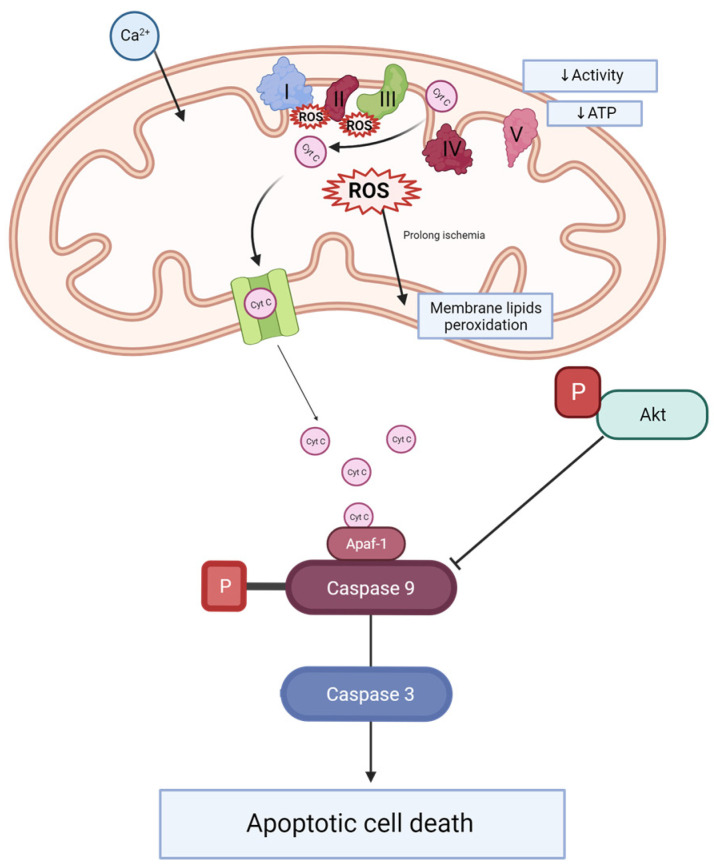
Scheme of caspase activation and Akt signaling during kidney injury. During ischemia-reperfusion, an increase in mitochondrial calcium is observed. This leads to electron transporter chain hyperactivity and increased ROS production. ROS causes the peroxidation of membrane lipids. Changes of ETC activity and mitochondrial membrane potential effects decrease complex V activity and ATP production. Additionally, cytochrome C leakage from mitochondrial space into the cytosol activates caspase 9, which plays an important role in caspase 3 activation and apoptosis. Phosphorylated Akt inhibits caspase 9 activation. Ca 2+—calcium, ROS—reactive oxygen species, Cyt C—cytochrome C, I—complex I (NADH-ubiquinone oxidoreductase), II—complex II (succinate dehydrogenase), III—complex III (CoQ-cytochrome c reductase), IV—complex IV (cytochrome c oxidase), V—complex V (F1F0 ATP synthase), ATP—adenosine triphosphate, P—phosphate groups, Akt—AKT serine/threonine kinase 1, ETC—electron transporter chain, Apaf-1—apoptotic protease activating factor 1.

**Figure 7 ijms-24-14772-f007:**
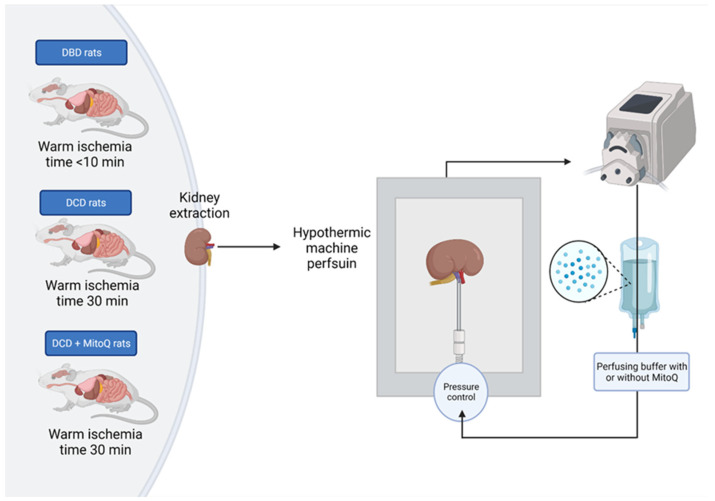
Scheme of the kidney preparation and perfusion. Right kidney was excised, flashed with pure buffer, and perfused for 22 h with or without MitoQ in the hypothermic condition (4 °C) with controlled pressure (50–120 mmHg).

**Table 1 ijms-24-14772-t001:** Information of reagents.

Name of the Product	Source	Identifier
Aprotinin	Roche (Basel, Switzerland)	10236624001
Bovine Serum Albumin	Sigma (Saint Louis, MO, USA)	A3294-100G
Buprenorphine	Orion Pharma, 0.3 mg/ml (Warsaw, Poland)	N/A
CaCl_2_	Sigma (Saint Louis, MO, USA)	C1016
Dithiothreitol	Roche (Basel, Switzerland)	10708984001
DMSO	Origen (Austin, Texas)	CP-50
EDTA	Chempur (Piekary Śląskie, Poland)	118798103
Glucose	Chempur (Piekary Śląskie, Poland)	114595600
KCl	Chempur (Piekary Śląskie, Poland)	117397402
KH_2_PO_4_	Sigma (Saint Louis, MO, USA)	P0662
Leupeptin	Sigma (Saint Louis, MO, USA)	L8511
MgSO_4_	Sigma (Saint Louis, MO, USA)	M7506
Mitoquinone	Abcam (Cambridge, United Kingdom)	ab285406
NaCl	POCH (Gliwice, Poland)	BA4121116
NaHCO_3_	Chempur (Piekary Śląskie, Poland)	428105306
DPBS	Gibco^TM^ (Thermo Fisher Scientific, Inc., Waltham, MA, USA)	A2644601
Sodium pentobarbital	Biowet, (133,3 mg + 26,7 mg)/ml (Puławy, Poland)	N/A
Soybean trypsin inhibitor	Sigma (Saint Louis, MO, USA)	P8340
Sucrose	Sigma (Saint Louis, MO, USA)	S0389
Tris-HCl	Roche (Basel, Switzerland)	10812846001
Triton X-100	Sigma (Saint Louis, MO, USA)	X100

## Data Availability

The data of this study will be available upon request from the reader.

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
