# Peer review of "Mitoquinone Alleviates Donation after Cardiac Death Kidney Injury during Hypothermic Machine Perfusion in Rat Model"

_ijms, 2023, doi:10.3390/ijms241914772_

Round 1

Reviewer 1 Report

Dear Author

Congratulation on your work. The manuscript presents a considerable amount of data worth being published. However, before acceptance for publication see some feedback that will allow to improve the manuscript.

Strength

1.       Good study design

Major Weakness

1.       There are major grammatical errors. Please see below the examples based on the abstract. Please seek a professional editing service (www.lannuo-editing.com / https://www.veritasedsci.com, etc.) and upload the editing certificate as a supplementary file.

·         Line 17-19: Background section; Use pleura form. Transplanted organs are subjected to harmful conditions through stopping blood flow, hypothermic storage of the graft and subsequent reperfusion.

·         Line 21: grammatical error: Should be “Antioxidants have gained interest….. “

·         Line 22: grammatical error: “properties to scavenging” – change to “ability to scavenge”

·         Line 22: “in the regenerative medicine” – should be “in regenerative medicine”

·         Line 23: first time mentioned use “a” instead of “the” ; “a key role in IRI”

·         Line 23: “a” is missing  - “strong” should be “a strong”

·         Line 25: “ following groups”, “a hypothermic”, and “ plural form “levels of kidney injury”.

·         Line 29: When a singular noun is mentioned for the first time with no other noun marker, use a (or an). Use “the” with any noun when the meaning is specific. Change to: We compared the activation of the apoptosis pathway (caspase 3 and 9), the concentration of phosphorylated Akt (pAkt) and the pAkt/total Akt ratio. MitoQ reduces KIM-1 concentration, total ROS/RNS and caspase levels. We observed a decrease in pAkt and the pAkt/total Akt ratio after drug administration.

·         Line 33: missing “the”, verb tense mistakes: impact - impacts, missing comm (always use comma after however), “an" goes before words that begin with vowels, “injury of” should be “injury to”. Change to: Conclusions: The length of warm ischemia time negatively impacts the graft condition. However, MitoQ added to the perfusing system as an on-pump therapy mitigates injury to the kidney before transplantation, inhibiting apoptosis and reducing ROS/RNS levels. We propose MitoQ as a potential drug for DCD graft preconditioning.

·         Beside vs in addition – In addition, is more common when 2 or more facts are mentioned in the same sentence. Maybe consider using “in addition” in line 55

·         Keep using abbreviation when already define, e.g., line 68 already defined in line 41; line 92 already defied in line 48.

·         Do not include abbreviate if they are not going to use used further in text, e.g., line 83; ” nonalcoholic fatty liver disease (NAFLD)

·         Line 91: change present perfect into past tense. You have already done the study.

2.       MitoQ concentration used in this study was based on what?

3.       Have author checked he toxicity of the MitoQ?

4.       Line 96 - verb tense mistakes: We showed elevated concentrations of KIM-

5.       1 in the DCD group (113.51 ± 63 ng/ml vs. 260.03 ± 40.72 ng/ml p=0.012 for DBD vs. DCD) compared to DBD.

6.       Line 118 ; verb tense mistakes ”Caspase 9 is initialing”  - should be “Caspase 9 initiates”

7.       In the figure description include the method, number of mice per experiment, technical replicates, and how p was calculated (e.g., “Data represents the mean ± SD (n = 3). **P < 0.01, ***P < 0.001, versus the control group”).

8.       Author mentioned the reduction of the caspase 3 and collagen 1A, however the results section did not show the Col1A results. “Mitoquinone administration reduces the expression of liver fibrosis proteins such as Col1A1 and the active form of caspase 3, which represents the executive apoptosis protease”. If author did not do the experiment but mentioned it, the reference should be provided.

9.       Author discusses the role of TGFb1 in Akt activation, with that there are no data to support it. If feasible include western blot for TGFb. “Transforming growth factor-β1 (TGF-β1) is proposed as an important inducer of kidney fibrosis and an activator of the Akt protein.

10.   Can author points where the data are presented : “CcO activity related to the duration of WIT”.

11.   Methods section is lacking details. Reader should be able to repeat the study following method section. In the current stage, it is impossible.

·         Total number of rats is missing. There is no information on the sex of animals.

·         Mitoqunone mesylate concentration used in the perfuse buffers is missing. Furthermore, author should provide the cytotoxicity/titration to show the dose used in the study is safe.

·         Assessment of kidney injury marker (KIM-1), no information on how much protein lysate was take, no information on the protein concentration.

·         Line 305:  “Then, a total of four times of centrifugation were performed to obtain the pellet of mitochondrial fraction”. The reader will have a hard time to follow what was done in this experiment. Provide the speed and time of centrifugation.

1.       Line 316: Author repeated the same massage in two following sentences : ” To determine the outer mitochondrial membrane integrity the activity of CcO was measured in the presence and absence of n-dodecyl β-D-maltoside. The outer mitochondrial membrane integrity is represented as mitochondrial cytochrome c oxidase activity in the presence and absence of the detergent n-dodecyl β-D-maltoside. Again, please provide step by step what was done to measure cytochrome C oxidase activity and outer mitochondrial membrane integrity – reader needs to know the volume, concentration, ratio, instrument name and run specification.

2.       Line 326. Section required details.

There are major grammatical errors.  Please seek a professional editing service. 

Author Response

Dear Reviewers,

Thank you very much for taking our manuscript into consideration for IJMS journal. We revised and corrected our manuscript following reviewers suggestions. The corrections are highlighted in red in improved manuscript. All corrections are described in word files. We believe that our paper is significantly improved and present high-quality data and study design. 

Reviewer 2 Report

In the paper “Mitoquinone mesylate plays a key role in maintaining the kidney organ in a good condition during transplantation” the authors evaluated the effect of Mitoquinone mesylate (MitoQ) - strong mitochondria-targeted antioxidant administrated directly to the perfusing buffer. I kindly request the authors to show additional data. I find the amount of data presented in this part of the study is insufficient to support many of the conclusions made in this work. Also the experimental is not detailed enough and description is vague. English language correction is needed; some sentences also are confusing.

As it stands, the paper is not suitable for publication in International Journal of Molecular Sciences.

The major concerns of this study are as follows:

1. Please exact information on all chemicals and reagents (MitoQ) including source and catalog numbers needs to be provided. Again different concentrations of MitoQ may need to be added.

2. Figure 1: * p<0.05 Sham vs IRI ?

3. Figure 4: Please unify the description in the figure and annotations. pAKT or p-AKT? Pakt/tAKT or pAkt/ total Akt? 

4. Figure 5A: There seems to be differences in it. I recommend that the authors reassess their analyses.

5. Authors should include expression of proliferation and apoptosis markers by IHC, WB or qPCR.

6. The number of rats used in each group of experiments has been included in study.

 English language correction is needed; some sentences also are confusing. 

Author Response

Dear Reviewers,

            Thank you very much for taking our manuscript into consideration for IJMS journal. We revised and corrected our manuscript following reviewers suggestions. The corrections are highlighted in red in the improved manuscript. All authors responces are described in the separate file in word. We believe that our paper is significantly improved and present high-quality data and study design.

Round 2

Reviewer 2 Report

The current version has addressed most previous comments from reviewerIt can be accepted at this stage. 

Author Response

RESPONSE TO REVIEWER 2

of “Mitoquinone plays a key role in maintaining the kidney organ in a good condition during transplantation.” by Anna Radajewska, Jakub Szyller, Anna Krzywonos-Zawadzka, Agnieszka Olejnik, Grzegorz Sawicki, Iwona Bil-Lula.

We would like to thank for taking your time to read and evaluate our manuscript again. We also  kindly thank for manuscript acceptance at this stage.

                                                                                        Authors
